# Characteristics of Compound Climate Extremes and Impacts in Singapore, 1985–2020

**Jianjun Yu ***, **Anupam Kumar, Kanhu Charan Pattnayak, Jeff Obbard**  **and Aurel Florian Moise**

Centre for Climate Research Singapore, Meteorological Service Singapore, National Environment Agency, Singapore 537054, Singapore
\* Correspondence: yu_jianjun@nea.gov.sg

**Abstract:** Compound weather and climate extremes have amplified impacts on natural and socioeconomic systems across the world, including Singapore. To better understand the spatial and temporal characteristics of compound climate extremes, including concurrent rainfall and wind speed, as well as dry and hot conditions, we analyzed long-term observations from 11 selected meteorological stations over the period 1985–2020. The results revealed that the north and northeastern parts of Singapore were focal points for both types of compound extremes, with a higher frequency of occurrence than the southwest of the island. Concurrent rainfall and wind speed extremes were the most prominent in December and January thanks to the northeast monsoon, while dry and hot extremes were distributed mainly in the inter-monsoon season, with peaks in March and April. A notable upward trend was also detected for mild and moderate levels of both compound climate extremes over time. According to our review of the impacts, Singapore has benefited from investments in enhanced water infrastructure; water resource availability was less affected; and flash floods were not proportionally related to the severity of climate extremes. The forests in the urban landscape of Singapore also exhibit resilience to drought.

**Keywords:** compound climate extremes; rainfall; wind speed; dry and hot; Singapore; climate impacts



## 1. Introduction

Compound weather and climate extremes, i.e., two or more extreme events simultaneously or successively occurring, often lead to amplified impacts on natural and socioeconomic systems [1]. For example, compound rainfall and wind extremes driven by convective weather systems can cause a series of impacts, such as building damage, uprooted trees, and flash floods. In coastal areas, such impacts, induced by tropical cyclones, can amplify the magnitude of inland flooding due to the restricted runoff of heavy rainfall by elevated coastal seawater levels [2]. The simultaneous occurrence of drought and hot extremes has been shown to increase rates of tree mortality across Europe and wildfire danger in Australia [3,4]. Global warming and the projected alteration of the hydrological cycle can be expected to exacerbate the risks of compound climate extremes in the coming decades [5]. Hence, a better understanding of the spatiotemporal variation, patterns, and trends of concurrent climate extremes is vital for policymakers and stakeholders to better assess, design, plan, and adapt to associated negative impacts.

There have been numerous studies reported, globally and regionally, that have analyzed the concurrence of climate extremes, as related to dry and wet conditions. Studies on compound dry and hot extremes surged in the recent decade after Europe experienced an intense heatwave that was accompanied by a severe drought condition in 2003 [6]. These studies accounted for the different components of concurrent dry and hot extremes, including spatial variability, frequency, and severity [7]. Globally, a significant increase in the severity of concurrent dry and hot extremes has been observed in many regions, including the western US, northern South America, Western Europe, Africa, Western Asia,

Southeast Asia, South India, northeast China, and eastern Australia [8]. These observations were broadly consistent with the findings from many other, more-recent regional studies, such as the wide and frequently occurring events in Europe [9] and the observed increases in widespread and persistent events in China [10]. Compound dry and hot extremes have also substantially increased in likelihood in the US [11] and in India [12] over the past several decades.

Studies on the concomitant occurrence of precipitation and high-wind events are relatively few, but they are important in the context of insurance to calculate loss and damage. Martius et al. quantified the global occurrence of compound precipitation and wind extremes, with and without spatial and temporal shifts, using ERA-interim reanalysis data from 1979–2012. It was found that a higher frequency of concurrent extremes occurred in coastal regions and in those areas suffering from frequent tropical cyclones [13]. According to observational data, Zhang et al. drew a similar conclusion for China, where concurrent extremes were more likely to occur in southern and southeast coastal areas, with an upward trend of mild and moderate events after 2011 [14]. A comprehensive global-scale study was reported by Ridder et al. [15], which analyzed 27 hazard pairs to identify hot spots, seasonality, and the most dominant hazard pairs in each region.

Despite the high vulnerability of Southeast Asia to compound climate extremes, there is a dearth of empirical studies on this topic, particularly at the urban scale. This gap is partly due to a lack of sufficient observational networks in the region. A comprehensive understanding of concurrent climate extremes is crucial for the developed, high-density population city-state of Singapore, to support the nation's sustainable development and future resilience to climate change.

An analysis of univariate climate extremes in Singapore has been previously reported. Li et al. analyzed extreme dry and wet conditions on the basis of using rainfall observations from 1982 to 2013 and showed fewer rainfall extremes, in both frequency and intensity, while the length of extreme dry spells became longer [16,17]. Beck et al. observed an increase in rainfall totals during the period 1980–2010 and observed a rapid increase in daily rainfall extremes compared with hourly rainfall [18]. Rahardjo et al. identified that the 5-day antecedent rainfall exhibited a different spatial pattern with that of annual rainfall, with a larger amount of rainfall in the northern and northeast parts of Singapore [19]. For temperature extremes, Jiang et al. observed a general increasing trend for most of the indicators, according to the Expert Team on Climate Change Detection and Indices (ETCCDI), except for the number of cool days and cool nights and the diurnal temperature range, which showed a downward trend [20]. While univariate climate extreme analyses on rainfall or temperature have provided valuable information for urban planning in Singapore, an investigation into compound climate extremes has been overlooked.

Using the historical observational data from the network of meteorological observation stations in Singapore, the main objectives of this study are as follows: (i) to investigate spatial variation of concurrent rainfall and wind speed extremes on daily and 3-h temporal scales, as well as the concurrence of dry and hot extremes, under different threshold combinations; (ii) to examine the inter- and intra-annual variability of the aforementioned concurrent events and identify any temporal trends; and (iii) to review local climate impacts by using available information. The findings of this study advance our understanding of the observed compound climate extremes in Singapore over time and inform evidence-based policies and strategies that aim to create a more climate-resilient Singapore.

## 2. Materials and Methods

### 2.1. Study Area and Data

Singapore, a tropical island city-state country with an area of around 729 km$^2$, lies at the southern tip of Peninsular Malaysia. Being situated 1° north from the equator, Singapore can be readily classified as having an equatorial climate, as characterized by abundant rainfall, high temperatures, and high humidity throughout the year. According to the climatological records maintained by the Meteorological Service Singapore (MSS) [21],

the long-term mean annual rainfall total during the period 1981–2010 was 2165.9 mm, with a higher amount of rainfall in the northern and western parts of the island, decreasing toward the east. The average number of dry days in a year is 198. The northeast monsoon (from December to early March) and southwest monsoon (from June to September) seasons introduce two extreme weather systems: monsoon surges and Sumatra squalls, both of which lead to short-duration events of heavy rainfall, accompanied by strong wind gusts. The most prominent winds in Singapore are from the northeast and the south, and they are generally light, with a mean wind speed of less than 2.5 m/s. Wind gusts with speeds of 10 m/s or more have been observed during the northeast monsoon surge and during thunderstorms. The daily mean temperature in Singapore ranges from a minimum of 23 °C to 25 °C to a maximum of 31 °C to 33 °C. Influenced by a complex urban topography and the urban heat island effect, the difference in microclimate results in a significant spatial and temporal variation of ambient temperatures across Singapore by as much as 3 °C during daytime hours and 7 °C at night [22,23].

The long-term data sets for hourly rainfall, surface air temperature, and wind speed in Singapore were sourced from MSS. We selected a total of 11 meteorological stations for our study. All three climate variables were collected at these stations for a sufficiently long period, and they are spatially distributed across the island (Figure 1). Of the 11 stations, the observations from four stations (i.e., S6, S23, S24, and S25) span 36 years, from 1985 to 2020, and those from the rest cover the period 2011–2020. For each station, the hourly data sets were processed to produce daily rainfall total ($R_d$), 3-h rainfall total ($R_{3h}$), daily mean wind speed ($W_d$), 3-h mean wind speed ($W_{3h}$), and daily maximum surface air temperature ($T_m$), for further analyses. For a detailed description of the meteorological data sets in Singapore, refer to [16,17,20].

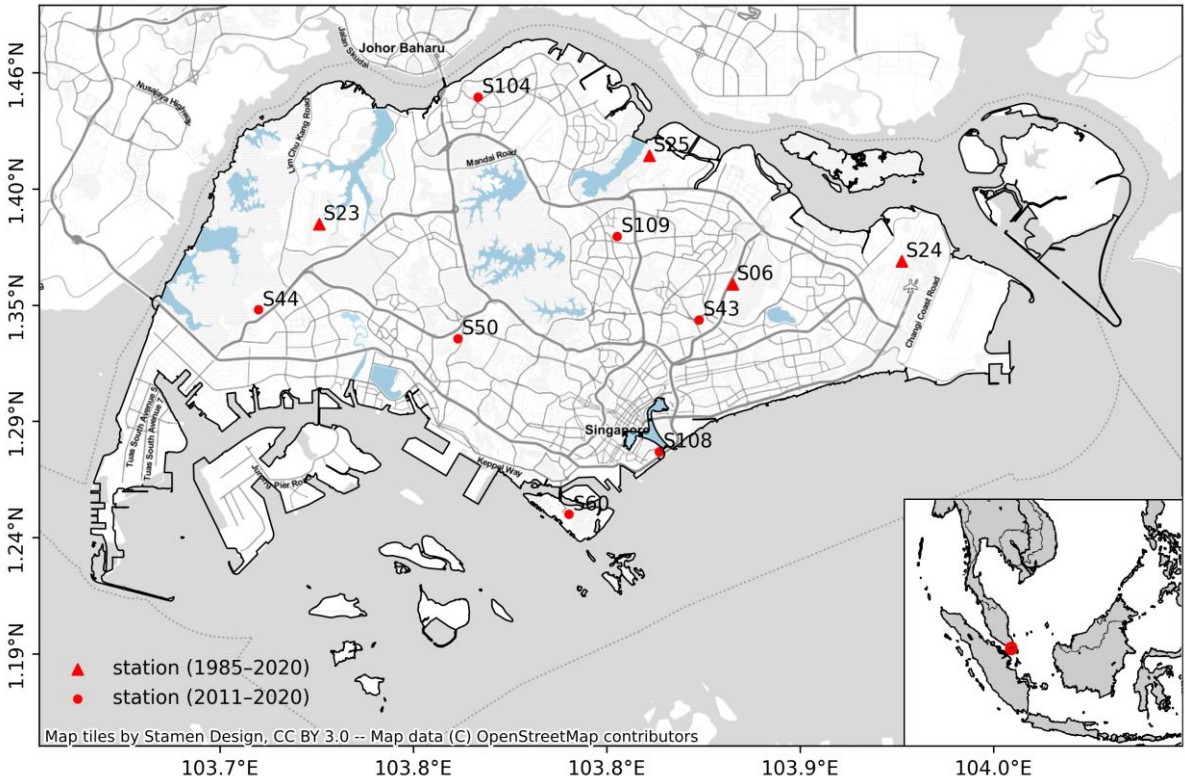

**Figure 1.** Study area and the location of the 11 selected meteorological stations in Singapore.

## 2.2. Definition of Compound Climate Extremes

In this study, we examined the spatial and temporal characteristics of concurrent rainfall and wind speed extremes, at the daily ($CRWE_d$) and 3-h ($CRWE_{3h}$) time scales, as

well as the concurrence of dry and hot extremes (*CDHE*). These variables are potentially relevant to the impacts of water shortage, flash floods, tree failure (i.e., tree uprooting or branches dropping), and vegetation fires in Singapore. The $CRWE_d$ (or $CRWE_{3h}$) is defined as the extreme rainfall and wind speed events occurring at the same station on the same day (or during the same 3-h period) or that are displaced in time by 1 day (or 3 h). This temporal relaxation takes into account the condition that both extremes associated with the same weather system but which occur with a time lag and/or cross the midnight time boundary [13]. The *CDHE* is defined as the concurrence of consecutive dry days (*CDD*) and extreme daily maximum surface air temperature at the same station. A day with a total rainfall ($R_d$) of less than 1 mm is considered a dry day.

The percentile-based approach has been widely adopted to define extremes in climatology [24]. Table 1 lists the statistics and percentiles of climate variables $R_d$, $R_{3h}$, $W_d$, $W_{3h}$, $T_m$, and *CDD*. The percentiles were calculated, respectively, on the basis of the data sets collected at meteorological stations S6, S23, S24, and S25 from 1985 to 2020. The 36-year time series captures decadal climate variabilities in Singapore and represents sufficient sample data points to ensure statistical robustness. A combination of different levels of rainfall and wind speed extremes (i.e., the 90th-, 95th-, and 98th-percentile thresholds) was selected for studying $CRWE_d$ and $CRWE_{3h}$. In some cases, the severity of extremes, namely mild, moderate, or severe, was used as an alternative to the thresholds at the 90th, 95th, and 98th percentiles, respectively. An analysis of *CDHE* was based on the combination of 1, 3, and 7 *CDD* and $T_m$ with thresholds at the 90th, 95th, and 98th percentiles.

**Table 1.** Statistics and percentile-based thresholds of climate variables.

| Climate Variable | Mean | Standard Deviation | 90th Percentile | 95th Percentile | 98th Percentile |
|---|---|---|---|---|---|
| Daily rainfall total $R_d$ (mm) | 6.7 | 15.2 | 22.2 | 36.1 | 56.0 |
| 3-h rainfall total $R_{3h}$ (mm) * | 6.6 | 11.6 | 19.4 | 29.9 | 44.6 |
| Daily mean wind speed $W_d$ (m/s) | 2.2 | 1.1 | 3.7 | 4.3 | 5.0 |
| 3-h mean wind speed $W_{3h}$ (m/s) | 2.1 | 1.6 | 4.3 | 5.1 | 6.0 |
| Daily maximum air temperature $T_m$ (°C) | 31.4 | 1.7 | 33.4 | 34.1 | 34.4 |
| Consecutive drying days *CDD* ** | 3.7 | 2.9 | 7 | 9 | 12 |

\* The statistics and percentiles of $R_{3h}$ are calculated on the basis of the data set that excludes the zero values.
\*\* The statistics and percentiles of *CDD* are determined on the basis of daily rainfall averaged across the stations.

There is no distinct wet or dry season in humid Singapore, where a greater diurnal variation of rainfall during the day [18] and vast spatial differences over land and coastal areas are observed [25]. Selecting relatively lower thresholds of dry conditions for the mean and 90th percentile (i.e., *CDD* of 3 and 7) enabled a spatial comparison of the dry and hot extremes in the city-state of Singapore. In addition, the combination of 1 *CDD* with $T_m$ was used as a proxy of hot extremes in Singapore, given that the prevalence of heat stress is a growing concern as a result of global warming.

To examine the spatial variability of compound climate events, we applied the same thresholds to each meteorological station to calculate the frequency (i.e., the number of events). Temporally, we counted the total number of events that occurred during each year to determine any temporal patterns for Singapore, followed by the application of the Mann–Kendall trend test [26]. The number was tallied for each event that occurred at any station. However, for those that simultaneously occurred in multiple stations, the events were counted only once. Furthermore, the percentage of climate extreme events that occurred in each month over the total time period was analyzed to examine intra-annual variability.

## 3. Results

### 3.1. Compound Rainfall and Wind Speed Extremes

As shown in Figure 2, the $CRWE_d$ frequency showed a similar spatial pattern with different combinations of rainfall and wind speed thresholds. We observed a higher frequency

in the northern and northeast parts of Singapore, decreasing toward the southwest. The $CRWE_d$ frequency was greater than 3 days per year in the north, especially along the coast, when both the rainfall and the wind speed exceeded their respective 90th-percentile thresholds (Figure 2a). The frequency was 0.8–1.3 (Figure 2e) and 0.1–0.2 (Figure 2i) days/year, respectively, when both thresholds were above the 95th and 98th percentiles. This implies that the return period of a $CRWE_d$ event at the 98th-percentile threshold was equivalent to 5–10 years. In contrast, the $CRWE_d$ frequency at the southwest stations (i.e., S44 and S50) was less than 0.1 days/year when the wind speed threshold was at the 90th percentile, when combined with any rainfall threshold (Figure 2a–c). There was no compound event observed when the wind speed threshold increased to the 95th or 98th percentiles (Figure 2d,g). The $CRWE_d$ frequency more rapidly decreased with an increase in the wind speed threshold at a given level of rainfall threshold when compared to the combinations in reverse. This indicates that extreme wind speed events have more influence on the spatial patterns of the concurrent extremes. Hence, those areas that experience high wind speeds tend to have more-frequent concurrent extreme events.

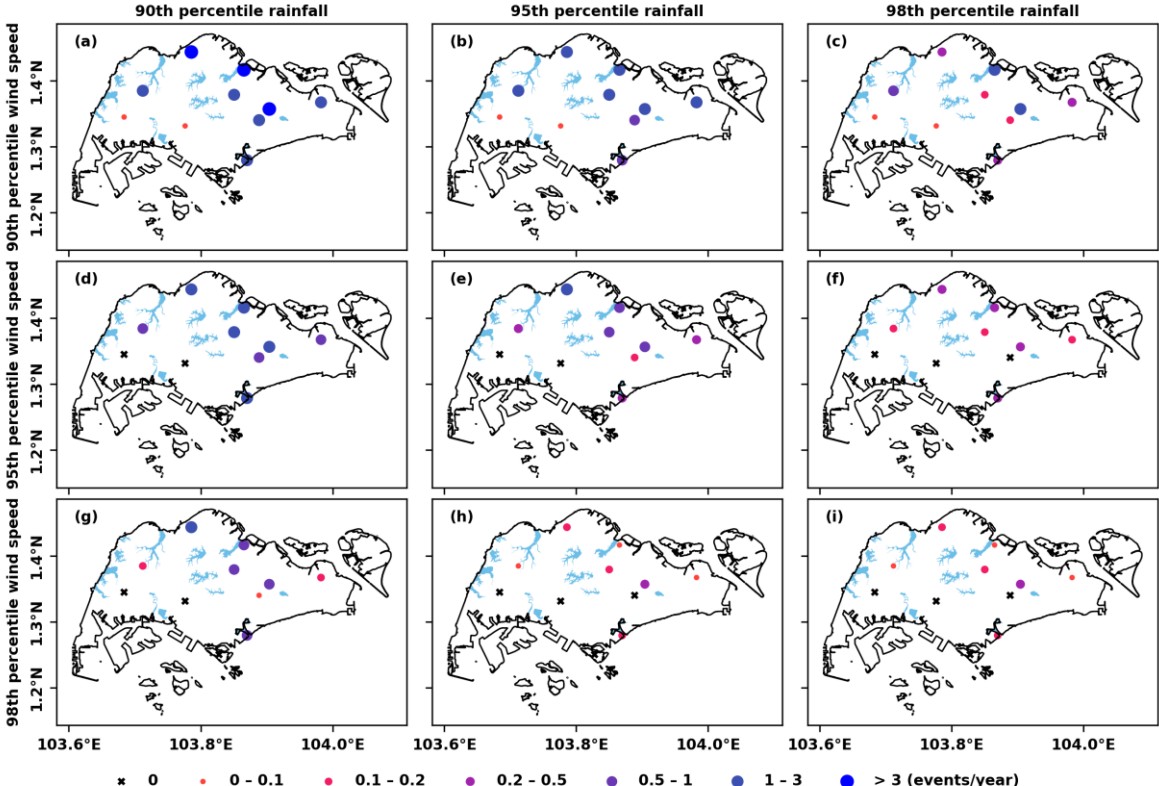

**Figure 2.** Frequency (events/year) of concurrent daily rainfall and wind speed extremes across Singapore. Rows represent daily mean wind speed thresholds (90th, 95th, and 98th percentiles). Columns represent daily rainfall total thresholds (90th, 95th, and 98th percentiles).

The spatial pattern of $CRWE_{3h}$ frequency (Figure 3) was basically similar to that of $CRWE_d$ (Figure 2), showing a northeast–southwest gradient. According to a comparison between it and $CRWE_d$, the $CRWE_{3h}$ frequency slightly increased to greater than 3.0 events/year in the east (i.e., S24) and northwest (i.e., S23) of Singapore and to 0.2–0.5 events/year in the southwest of the island when both 3-h rainfall and wind speed exceeded the 90th-percentile thresholds (Figure 3a). The high 3-h wind speed confined the distribution of $CRWE_{3h}$ events mainly to the northeast, with a frequency of 0.1 events/year (Figure 3i), which is equivalent to a 10-year return period. The extreme weather systems (i.e., cold surges, Sumatra squalls, and convective thunderstorms) in Singapore brought in high-intensity rainfall over a 3-h period, which significantly contributed to the amount of

daily rainfall during an extremely wet day. Hence, the frequency of extreme 3-h rainfall events was at the same level as that of an extreme daily rainfall event, especially at the thresholds of 95th and 98th percentiles (Figures S1 and S2). Although the extreme 3-h wind speed frequency increased, the frequency of $CRWE_{3h}$ was limited by that of extreme 3-h rainfall, resulting in a similar level of frequency for $CRWE_d$. In the station on the southern island (i.e., S60 on Sentosa), no extreme event was observed for either $CRWE_d$ (Figure 2) or $CRWE_{3h}$ (Figure 3b–f). This indicates a milder coastal climatic condition with a relative low intensity of daily and 3-h rainfall and wind speed in Singapore's southern islands. In general, the frequency of rainfall extreme events determines the maginitude of the frequency of compound extreme events, while that of wind speed extremes dominates the spatial varibility.

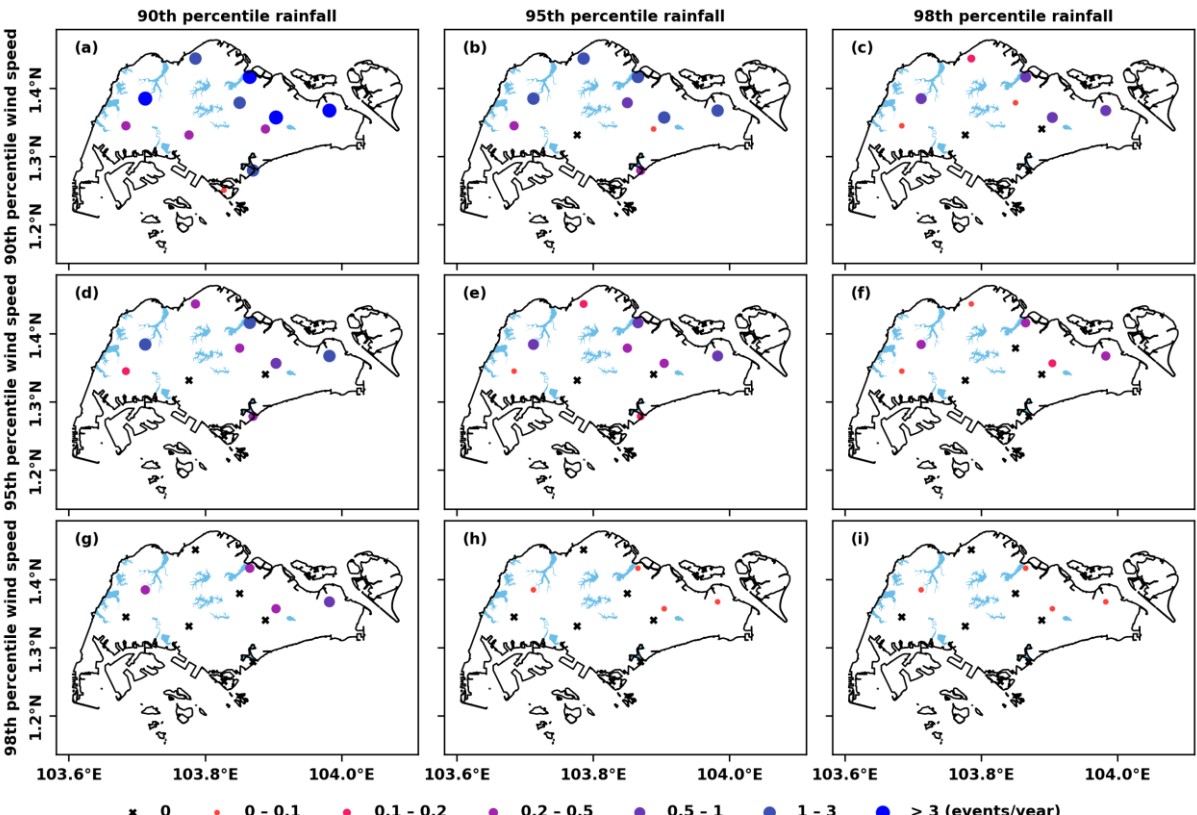

**Figure 3.** Frequency (events/year) of concurrent 3-h rainfall and wind speed extremes across Singapore. Rows represent 3-h mean wind speed thresholds (90th, 95th, and 98th percentiles). Columns represent 3-h rainfall total thresholds (90th, 95th, and 98th percentiles).

Figure 4 shows the interannual variation of $CRWE_d$ and $CRWE_{3h}$ events for a different combination of thresholds for Singapore during the period 1985–2020. For the 90th percentile of rainfall threshold, the annual number of $CRWE_d$ events varied from 2 to 10 during the 1985–2009 period when the wind speed above the 90th-percentile threshold (Figure 4a), occurred for only a few years at a number above 10 events and at a maximum number of 14 events. The range was from 0–10 when the wind speed threshold increased to the 95th percentile. The concurrent events increased after 2010, to a level around 10–30 every year for both wind speed thresholds. An upward trend was detected with a rate of 0.2 and 0.14 events/year, separately. The temporal pattern for the 95th-percentile rainfall threshold was basically similar (Figure 4b), with fewer concurrent events (i.e., mainly fewer than 10) every year and a lower upward trend (i.e., increasing by fewer than 0.1 events/year). There was no significant trend detected when rainfall exceeded the 98th-percentile threshold.

However, it was obvious that severe concurrent events occurred more frequently after the year 2000, and they have occurred every year in the past 5 years (Figure 4c).

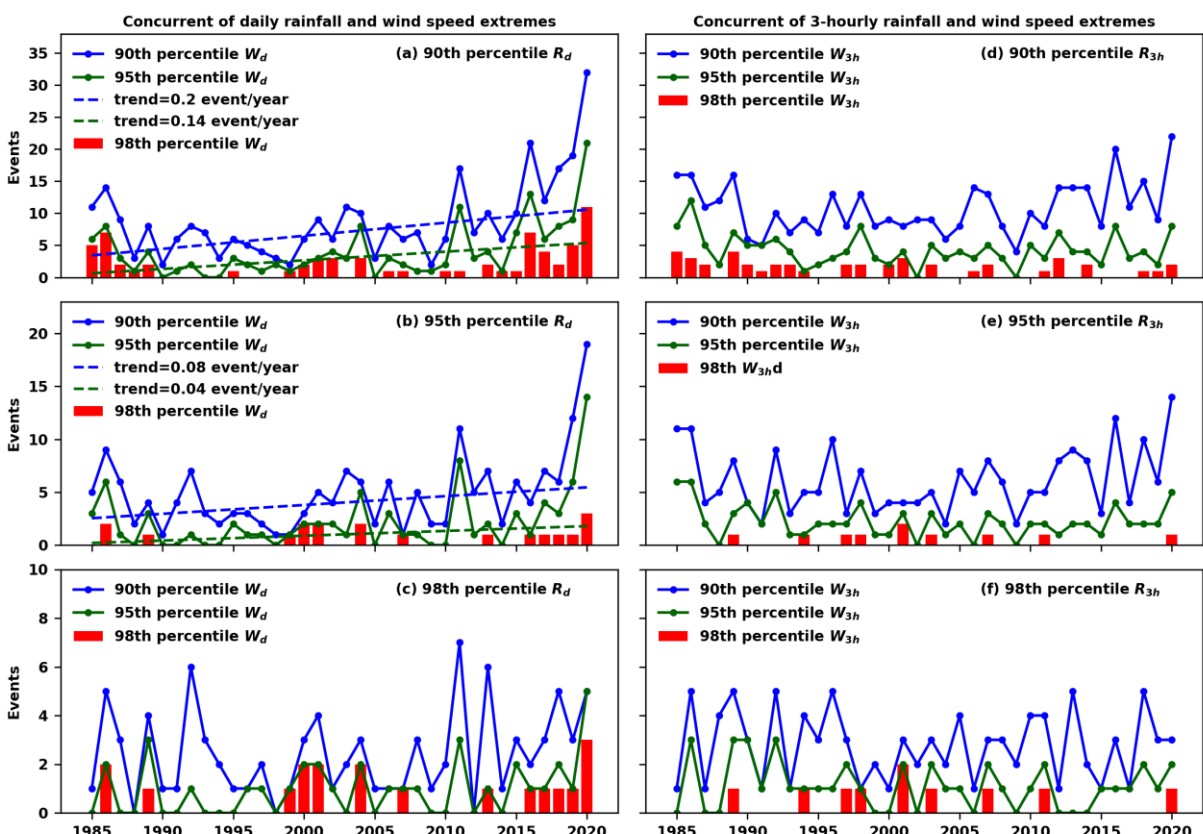

**Figure 4.** Interannual variability (number of events per year) of concurrent daily (**a**,**c**,**e**) and 3-h (**b**,**d**,**f**) rainfall and wind speed extremes in Singapore during the period 1985–2020 under different combinations of percentile-based thresholds.

The annual number of $CRWE_{3h}$ events varied in a similar range to that of $CRWE_d$ for the combinations of the 90th- and 95th-percentile thresholds, but without showing significant trends during the 1985–2020 period (Figure 4d,e). The number of concurrent events has slightly increased in recent years, where, for example, more years are observed to have over 10 events/year when the rainfall threshold was above the 90th or 95th percentile. In contrast to $CRWE_d$, the occurrence of severe $CRWE_{3h}$ events was seldom observed after the year 2000, and such events have happened only twice, i.e., in 2011 and in 2020 (Figure 4f). More-frequent severe concurrent events were observed in the period from 1997 to 2003. Generally, the concurrence of short-term (i.e., 3 h) rainfall and wind speed does not show an increasing signal. However, the increasing concurrence of moderate extremes of daily rainfall and wind speed is remarkable. It is due mainly to the wind speed increase in short durations in the northern part of Singapore (i.e., S104, S25, and S109), resulting in the increase in daily wind speed, whereas the rainfall intensity remains at the same level. This can be attributed to stronger wind effects during the longer northeast monsoon surges in recent years. Although severe concurrent events have occurred only in certain years, the consecutive occurrence in recent years is an indicator of climatic change.

Figure 5 shows the intra-annual variability of $CRWE_d$ and $CRWE_{3h}$ in terms of the percentage of the concurrent events that occurred in each month over the total period, 1985–2020. The $CRWE_d$ exhibits a consistent intra-annual characteristic under a combination of thresholds with different percentiles. As expected, the northeast monsoon brought in more events of rainfall and high-speed wind gusts, leading to a peak of $CRWE_d$ in the months of December and January, which accounted for around 50% of the total number of

events for mild and moderate extremes (Figure 5a–c). This percentage was even higher, up to 70%, for the wind speed threshold above the 98th percentile (Figure 5b,c). This implies that severe extreme events were more likely to occur in the months of December and January, which are dominated by high-speed wind conditions. The $CRWE_{3h}$ was distributed relatively evenly in each month when the wind speed threshold was above the 90th or 95th percentile (Figure 5d–f), mainly varying from 5% to 15%. In total, 10 severe extreme events occurred during the period 1985–2020 (Figure 4d,f), and three events occurred in December, accounting for 30% of the total (Figure 5e,f).

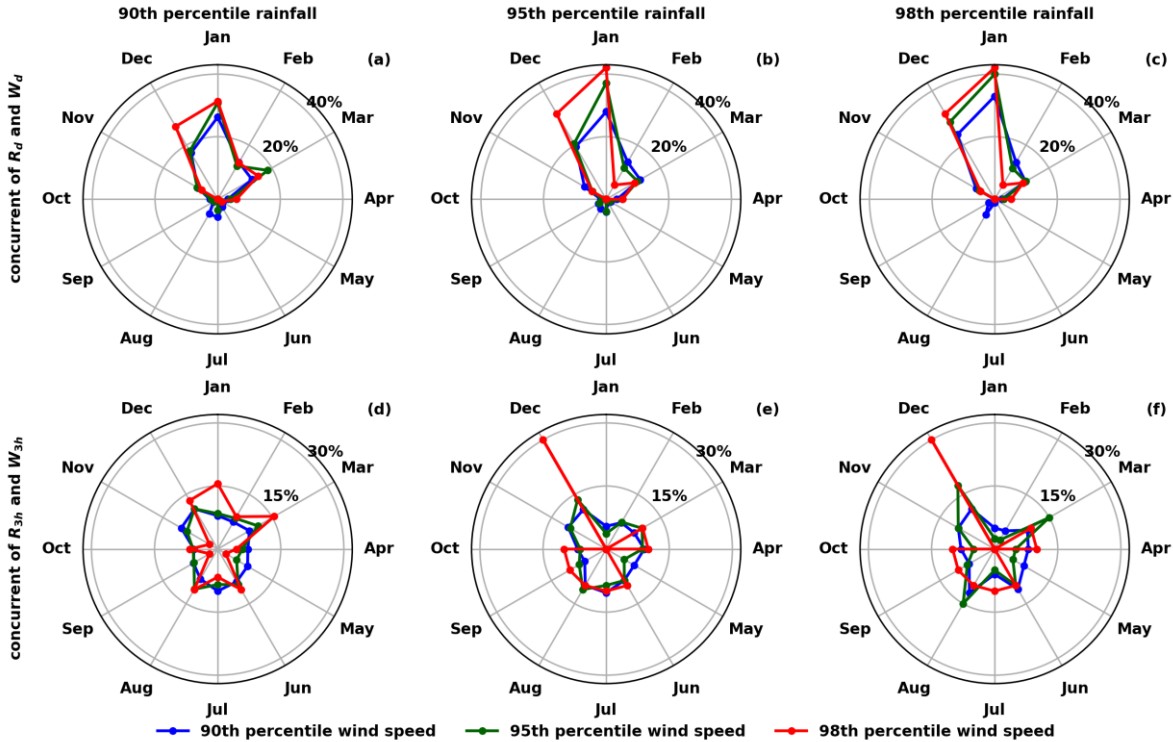

**Figure 5.** Intra-annual variability (number of events in the month relative to the total) of concurrent daily (**a**–**c**) and 3-h (**d**–**f**) rainfall and wind speed extremes under different combinations of percentile-based thresholds.

### 3.2. Compound Dry and Hot Extremes

The *CDHE* events with 1 *CDD* represent different levels of hot extremes on dry days. As shown in Figure 6, the central eastern parts of Singapore appear to be hot spots, where many stations were observed to have, on average, 20–40 hot days each year when the $T_m$ threshold was above the 90th percentile (Figure 6a). This frequency was reduced to around 10–20 days/year (Figure 6b) and 3–10 days/year (Figure 6c) when the $T_m$ exceeded the 95th- and 98th-percentile thresholds, respectively. The frequency of hot days was relatively lower in the western and southern parts of Singapore, which exhibited a milder climate, with 5–8 hot days/year decreasing to 0.2–0.5 days/year when the $T_m$ threshold increased from the 90th to the 98th percentile.

Singapore is more likely to experience prolonged consecutive dry days in the northeast than in the southwest of the island. At the 90th-percentile $T_m$ threshold, the frequency of *CDHE* events in the northeast was 0.4 to 1.2 events/year for more than 3 *CDD* (Figure 6d) and 0.1 to 0.4 events/year for more than 7 *CDD* (Figure 6g). The temperature, as influenced by the urban microclimate, appears to shape the spatial distribution of *CDHE* hot spots. The severe *CDHE* events (i.e., 7 *CDD* in combination with 98th-percentile $T_m$ threshold) were observed at stations S06 (i.e., Paya Lebar) and S23 (i.e., Tengah), with a frequency of around 0.05 events/year (Figure 6i), which was equivalent to a 20-year return period event.

This result was expected for S06, because it is located in one of the urban heat island hot spots of Singapore according to remote-sensing studies [27], where the observed annual mean temperature is around 1 °C higher than the average across Singapore. For S23, the severe *CDHE* events can likely be attributed to the rapid urbanization in recent years. The annual mean temperature observed in S23 quickly increased after 2016. In general, the northeast of Singapore is more likely to suffer compound climate extreme events, including a concurrent of rainfall, an elevated wind speed, and drier and hotter conditions. However, the microclimate was relatively mild in the southwest and southern islands of Singapore.

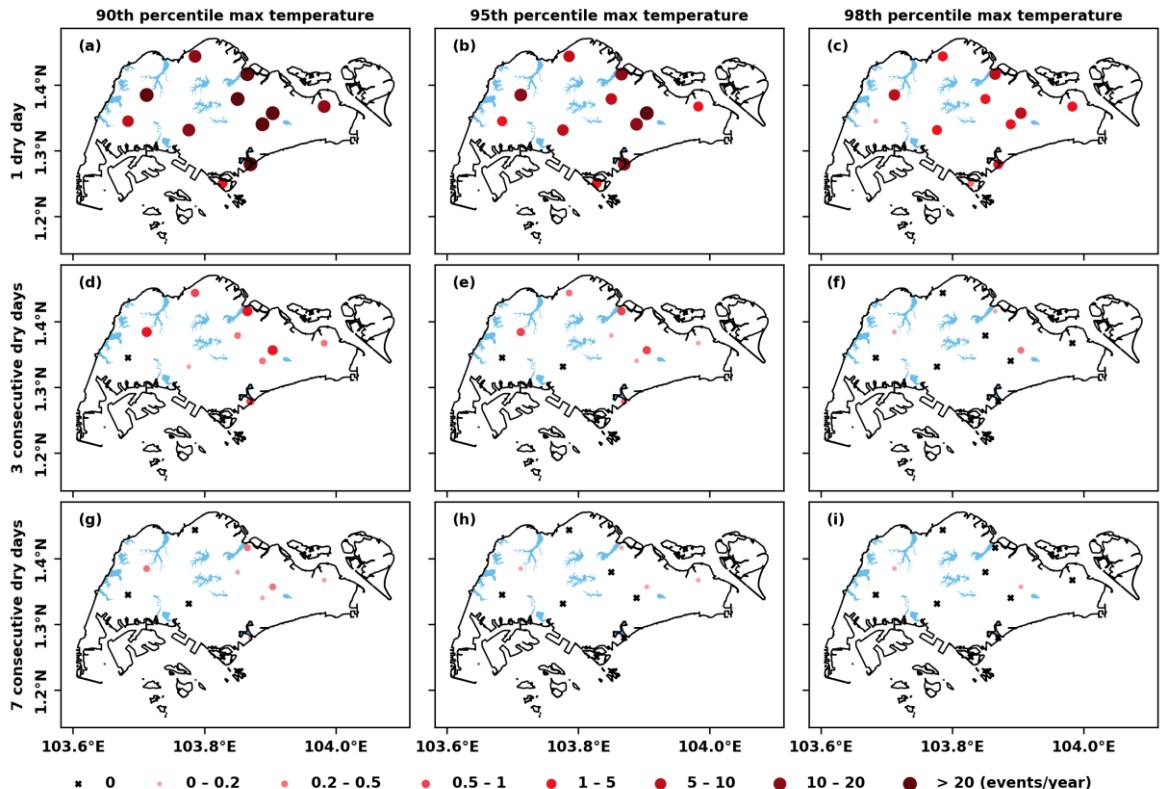

**Figure 6.** Frequency (events/year) of concurrent dry and hot extremes across Singapore. Rows represent different number of consecutive dry days (1, 3, and 7 days). Columns represent daily maximum air temperature thresholds (90th, 95th, and 98th percentiles).

Figure 7 shows the intra- and interannual variability of *CDHE* events under the scenarios represented by combing different numbers of *CDD* and $T_m$ thresholds. It is clear that the incidence of hot days showed a significant upward trend during the 1985–2020 period, with a slope of 1.8 days/year for the 90th-percentile $T_m$ threshold, 1.4 days/year for the 95th-percentile $T_m$ threshold, and 0.7 days/year for the 98th-percentile $T_m$ threshold (Figure 7a). A similar trend is also apparent for *CDHE* events with more than 3 *CDD* combining with mild and moderate $T_m$ thresholds, showing an annual increase of 0.1 and 0.05 in the number of events, respectively (Figure 7c). Although there was no significant trend detected from 1985 to 2020, the concurrence of *CDHE* events with more than 3 *CDD* and above the 98th-percentile $T_m$ threshold was more frequent after 2010 (i.e., it occurred every year, except in 2015), compared with that of only four times observed during the 1985–2009 period (Figure 7c). The severe *CDHE* events for more than 7 *CDD* and at the 98th-percentile $T_m$ threshold were seldom observed, and they occurred only twice, in 1998 and in 2016 (Figure 7e). Overall, the increasing frequency of extreme temperature events in Singapore, especially when accompanied with dry conditions, may be of particular concern in climate adaptation planning, where signals of more-frequent moderate concurrent events have been observed in more-recent years.

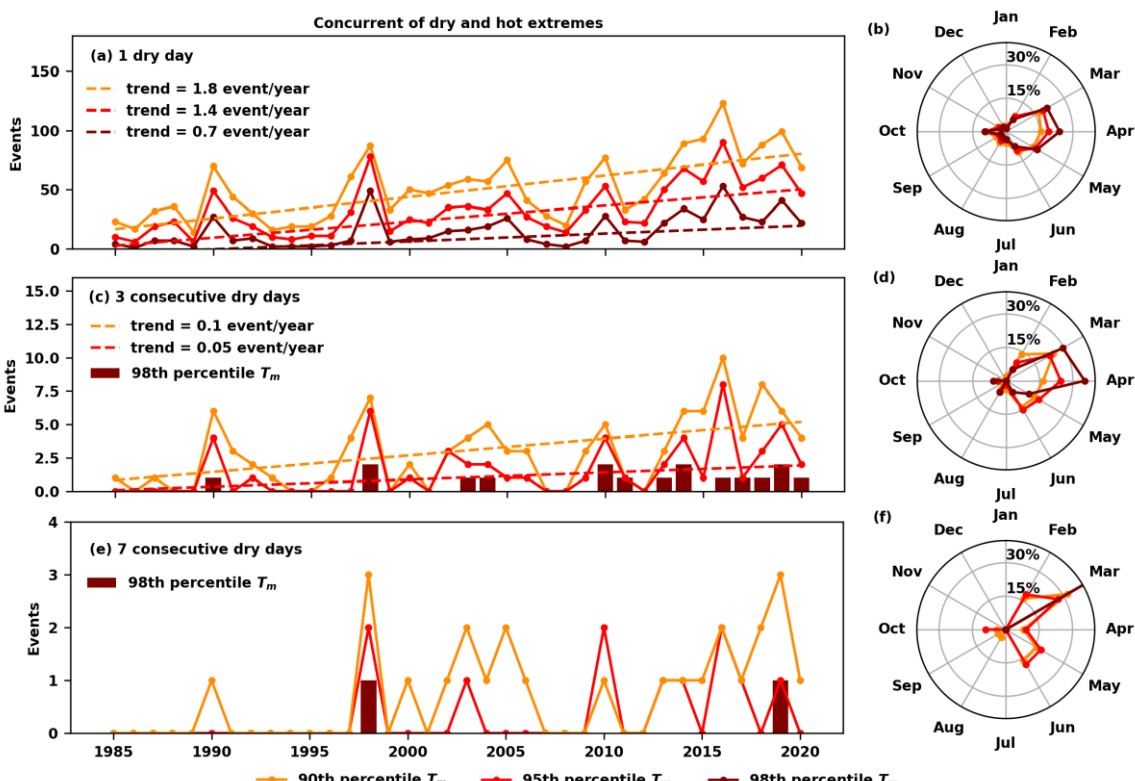

**Figure 7.** Inter- (number of events per year) (**a**,**c**,**e**) and intra-annual (number of events in the month relative to total) (**b**,**d**,**f**) variability of concurrent dry and hot extremes under different combinations of consecutive dry days and percentile-based daily maximum air temperature thresholds.

Monthly, the *CDHE* events were distributed mainly in the intermonsoon season (from February to June) and from August to October, with peaks occurring in March and April, when the climate was relatively dry and warm (Figure 7b,d,f). For 1 *CDD*, the percentage of *CDHE* events was around 32% and increased to around 38% and 43% (Figure 7b) when the $T_m$ threshold increased from the 90th percentile to the 95th and 98th percentiles, respectively. It shows that high temperatures after the monsoon season shape the concurrence of dry days and hot days. This characteristic could also be observed when the *CDD* was more than 3 days. The occurrence of corresponding *CDHE* events accounted for 41%, 49%, and 64% of events in March and April for different levels of $T_m$ thresholds (Figure 7d). There were 25 events observed during 1985–2020 for *CDHE* events with more than 7 *CDD*. In combination with mild and moderate temperature extremes, those events occurred mainly from February to June and from August to October. However, given the relatively high temperature in the intermonsoon season, the severe *CDHE* events (i.e., with a 98th-percentile threshold) occurred only in March and only twice during the period 1985–2020 (Figure 7f).

## 4. Climate Impacts

Singapore, as a tropical island and as a developed city-state nation without a hinterland, has a natural lack of water resources and is therefore vulnerable to prolonged drought conditions. In its history, Singapore suffered drought events in the 1960s and early 1970s, which led to potable water rationing. Singapore's water resilience to meteorological drought has progressively improved over time owing to heavy investments in water technology, infrastructure, conservation, and management by Singapore's national water agency, the Public Utilities Board (PUB), to build what is referred to as the four national taps [28], which include (i) local catchment runoff that is stored in reservoirs; (ii) raw water

imported from Malaysia; (iii) desalinated water derived from sea water; and (iv) NEWater from recycled wastewater effluent.

Although the frequency of prolonged dry conditions increased in the 2010s (see Section 3.2), this did not result in increased municipal water consumption. Even in a 42-day dry spell in 2014, the water supply was maintained to supply a 5% increase in potable water demand. This demand was likely due to an increase in bathing activity and garden watering during the dry spell [29]. In part for the same reason, water consumption increased by around 9% to 154 L/capita/day in 2020. It was argued that a shift in human behavioral patterns during the COVID-19 pandemic, which resulted from working and learning from home, led to higher hygiene standards and more bathing activity [30]. This implies that future studies on water demand in Singapore should place more emphasis on demand change from the projected population increase and how climate change will affect water-use behavior rather than focus on water supply alone. However, improvements in the efficiency of the four national taps should not be neglected, because there is a limited understanding of climate impacts on patterns of energy consumption associated with the operation of water infrastructure. In the context of drying and warming events, this can be expected to affect not only potable water demand but also related energy costs for water supply. Optimizing the water–energy nexus in the context of a changing climate is therefore recommended.

The impact of droughts and heat waves on tropical forests has been widely reported in the literature in terms of increased tree mortality, reduced growth, changing species composition, the shifting of forest biomes, and increased wildfire risks [31]. There is increasing concern over the impact of warming and climate extremes on Singapore's natural habitats given that the remaining undisturbed areas are relatively small owing to extensive deforestation. Reports on the impact of drought are confined to phenological terms, but they show evidence of irregular flowering, mastering, and reproduction [32–35]. In fact, according to a field study conducted in Singapore's Bukit Timah Nature Reserve, the lowland tropical rainforest showed surprising resilience to drought [36]. Although the forest in Singapore island is in fragmented condition, there are few signs of forest degradation as a result of drought. However, drought-induced vegetation fires have a disproportionally large impact on Singapore [37]. According to the Singapore Civil Defence Force (SCDF) [38], Singapore experiences an average 500 vegetation fires every year, varying from around 200 to 900 cases. The peak seasons for fires tend to occur from January to March and from August to September, which are slightly in advance of the peaks for concurrent dry and hot extremes (see Section 3.2). It implies that the high-speed winds in the monsoon season amplify the risk of vegetation fires. However, the relationship between climatic variables (i.e., drought, temperature, wind speed, and humidity) and vegetation fire risks in an urban context is still not well understood.

Flash floods have been reported in cities across the world, and they also occur in Singapore. Specifically in Singapore, tree failure often coincides with flash flooding during heavy rainfall events that are accompanied by high wind gusts. Figure 8 shows the corresponding daily and 3-h rainfall and wind speed values for the flood and/or tree filature events reported in Singapore's national newspaper, the *Strait Times*. Here, we extended the flood inventory to 2020 by using the method presented by [39] and then applied the same approach to compile a tree failure inventory for the period 1985–2020 by conducting a web-based search for 'falling tree + Singapore'. There were 176 flood events and 108 tree failure events reported in the period, 25 events of which simultaneously occurred. Flash-flooding events showed a similar characteristic in response to daily and 3-h rainfall, with around 80% of the reported flooding above the 98th-percentile rainfall threshold and above 90% for the 90th-percentile threshold. This may be partly due to the short duration of the rainfall events in Singapore, which occur mainly within a 3-h period. Tree failures appeared to correlate with 3-h wind speed thresholds. About 38% and 80% of the reported tree failure events occurred above the 98th and 90th 3-h wind speed thresholds, respectively, compared to 12% and 37% for the same thresholds of daily



wind speed. In general, the 90th-percentile (or higher) threshold of 3-h rainfall and wind speed could be considered as a valid indicator to issue a public hazard warning. However, these climate extremes are not the only factors triggering flash flooding and tree failure hazards, and a better understanding of the susceptibility of the drainage system and a better understanding of tree vulnerability are needed to improve the validity of hazard warnings.

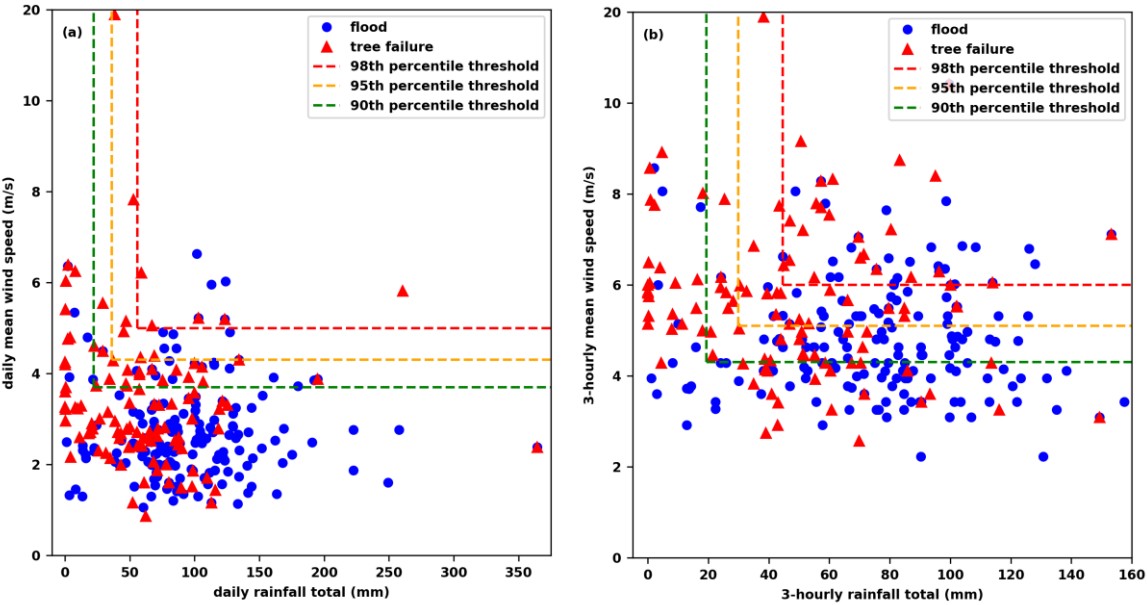

**Figure 8.** Daily (**a**) and 3-h (**b**) rainfall and wind speed corresponding to flood and tree failure events reported in the *Strait Times*.

## 5. Discussion and Conclusions

We empirically analyzed the frequency of compound climate extreme events in Singapore by using long-term observations from 11 selected meteorological stations. We observed a higher frequency (i.e., equivalent to a 10–20-year return period) of compound climate extremes in the north and northeastern parts of Singapore when compared to the south and southwest parts for concurrent rainfall and wind speed extremes, as well as for the concurrence of dry and hot extremes. This spatial variation indicates that investments, resources, and efforts should focus on those areas, to mitigate the potential of long-term impacts.

First, the mild and moderate compound extremes of daily rainfall and wind speed showed a slightly increasing trend, while no trend was detected for shorter durations at the 3-h time scale. In contrast, an observed trend of increasing warming and prolonged dry conditions was significant. This was, to some extent, consistent with the projected pattern of an increase in the frequency of heavy rainfall events and dry spells in Southeast Asia and Singapore [40,41]. Singapore's adaptation to future climate change may be best served by focusing on drying and warming scenarios, especially the rare, severe concurrent extremes. Furthermore, an altered hydrological cycle may result in cascading local impacts on water resource availability, corresponding energy consumption, and the resilience of urban greenery.

Second, the characteristics of concurrent climate extremes reported in this study could support an evaluation of how well high-resolution regional climate modeling simulates concurrent extremes. This would also help to elucidate the physical mechanisms represented in the climate models to support a more robust risk assessment of concurrent climate extremes.

Finally, we reviewed the climate impact in Singapore on the basis of using available data in the scientific literature, reports, and social media. Thanks to the advanced water supply infrastructure in Singapore, the water resource availability was not significantly

affected by climate extremes during the study period. Forests in the urban landscape of Singapore have, to date, also showed resilience to drought and heat. This is despite conservation biologists' stating that many tree species in tropical rainforests are already at the upper limit of their thermal tolerance threshold [42]. As expected, the occurrence of flash floods and tree failures was not proportionally related to the severity of climate extremes. The susceptibility and vulnerability of natural and/or human systems also plays an important role in the manifestation of climate impacts. Nevertheless, civil infrastructure and natural habitats that are exposed to a high frequency of concurrent climate extremes will increasingly need suitable protection and adaptation measures.

In this study, the frequency of concurrent climate extremes was based on counting the occurrence of events above certain thresholds. The frequency of percentile-based extremes was 'moderate' when the ability to apply such a definition relied on the availability of a sufficiently large number of observational data points. Our methodological approach did not reveal the extreme events in the far tail of the distribution curve, which may exist but which were not actually present in the historical records. Such events are expected to cause the most severe environmental and socioeconomic impacts. Hence, a sophisticated frequency analysis technique for multivariable distributions is a prerequisite for robust engineering design and adaptation planning.

In addition, for 7 of the 11 studied meteorological stations, only 10 years of meteorological observations were available, i.e., 2011–2020, which may have led to a bias in the findings. However, thanks to our method of using the observational data over the same 10-year period for all 11 stations, the spatial and temporal patterns of the concurrent climate extremes did not deviate. The results from this study serve as a valuable guide for adaptation planning and hazard preparedness in Singapore, where the influence of microclimate effects in a highly urbanized city-state should duly be considered.

**Supplementary Materials:** The following supporting information can be downloaded at https://www.mdpi.com/article/10.3390/cli11030058/s1.

**Author Contributions:** Conceptualization, J.Y. and A.K.; methodology, J.Y. and A.K.; formal analysis, J.Y. and A.K.; investigation, J.Y., A.K. and K.C.P.; data curation, J.Y., A.K. and K.C.P.; writing—original draft preparation, J.Y.; writing—review and editing, J.O., A.F.M. and A.K. All authors have read and agreed to the published version of the manuscript.

**Funding:** This research received no external funding.

**Data Availability Statement:** The data presented in this study are available on request from the corresponding author. The data are not publicly available, because of confidentiality agreements.

**Conflicts of Interest:** The authors declare no conflict of interest.

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
