# Peer review of "Characteristics of Compound Climate Extremes and Impacts in Singapore, 1985–2020"

_climate, doi:10.3390/cli11030058_

Round 1

Reviewer 1 Report

Attached

Reviewer 2 Report

The manuscript by Yu et al. focuses on analysis of compound extreme events in the Singapore area, with particular emphasis on co-incidence of periods of high precipitation sums and high wind speed, at daily and 3-hourly time scales. The authors provide an overview of spatial patterns of such events, as well as their temporal trends and seasonal specifics.  

The presented analysis is relative straightforward yet interesting, especially in the context of Singapore-specific aspects of extreme weather impacts and the related questions of climate change (which are also discussed in the text). The article is competently written and topically suitable for the Climate journal. I only have minor suggestions/questions regarding the results and their presentation (I leave it at the discretion of the authors whether they will incorporate any related changes to the revised manuscript).

Specific comments & questions:

(C1) An increasing trend in (especially) daily series is reported by the authors for the precipitation / wind speed compound extreme events (Sect. 3.1). It would be interesting, however, to also see if these changes in frequency of compound events are primarily due to trends in the individual series of wind speed or precipitation (for instance, if there were a higher amount of strong wind events recorded in the later part of the analysis period, translating into potentially higher number of the compound events), or if there was a change in the actual probability of the precipitation / wind extremes coinciding without major changes in their respective marginal distributions.

(C2) I find it curious that the absolute frequencies of the extreme compound events are (roughly) similar for the daily and 3-hourly data (as shown, e.g., in Fig. 4): Since there are 8 times more values in the 3-hourly series than in their daily counterparts (and thus 8 times more values exceeding the respective 90/95/98th percentile), I would expect the number of compound extreme events per year to also be higher by a similar factor (at least on average).

(C3) l. 276: “… according to remote sensing studies, …” – it would be useful to reference the relevant studies.

Round 2

Reviewer 1 Report

My comments and suggestions have been satisfactorily addressed by the authors. I consider that the manuscript can be published as it is.